# Mitochondrial Quality Control Mechanisms and the PHB (Prohibitin) Complex

**DOI:** 10.3390/cells7120238

**Published:** 2018-11-29

**Authors:** Blanca Hernando-Rodríguez, Marta Artal-Sanz

**Affiliations:** 1Andalusian Center for Developmental Biology, Consejo Superior de Investigaciones Científicas, Junta de Andalucía, Universidad Pablo de Olavide, 41013 Seville, Spain; bherrod@upo.es; 2Department of Molecular Biology and Biochemical Engineering, Universidad Pablo de Olavide, 41013 Seville, Spain

**Keywords:** mitochondria, stress response, mitophagy, mitochondrial dynamics, mitochondrial unfolded protein response (UPR^mt^), quality control, prohibitins, PHB complex, PHB-1, PHB-2

## Abstract

Mitochondrial functions are essential for life, critical for development, maintenance of stem cells, adaptation to physiological changes, responses to stress, and aging. The complexity of mitochondrial biogenesis requires coordinated nuclear and mitochondrial gene expression, owing to the need of stoichiometrically assemble the oxidative phosphorylation (OXPHOS) system for ATP production. It requires, in addition, the import of a large number of proteins from the cytosol to keep optimal mitochondrial function and metabolism. Moreover, mitochondria require lipid supply for membrane biogenesis, while it is itself essential for the synthesis of membrane lipids. To achieve mitochondrial homeostasis, multiple mechanisms of quality control have evolved to ensure that mitochondrial function meets cell, tissue, and organismal demands. Herein, we give an overview of mitochondrial mechanisms that are activated in response to stress, including mitochondrial dynamics, mitophagy and the mitochondrial unfolded protein response (UPR^mt^). We then discuss the role of these stress responses in aging, with particular focus on *Caenorhabditis elegans*. Finally, we review observations that point to the mitochondrial prohibitin (PHB) complex as a key player in mitochondrial homeostasis, being essential for mitochondrial biogenesis and degradation, and responding to mitochondrial stress. Understanding how mitochondria responds to stress and how such responses are regulated is pivotal to combat aging and disease.

## 1. Introduction

Mitochondria are essential organelles initially characterized by their role in energy metabolism, carrying out the tricarboxylic acid (TCA) cycle and oxidative phosphorylation (OXPHOS). However, nowadays we know that mitochondria are acting within an integrated network, communicating with the rest of cellular organelles [1]. Mitochondria are tightly connected with the endoplasmic reticulum (ER) [2] but also with lysosomes, lipid droplets and peroxisomes [3,4,5]. In particular, mitochondria are essential for Ca^2+^ homeostasis and Fe-S cluster biogenesis, two essential cellular processes [6,7]. Furthermore, mitochondria are involved in the synthesis of critical metabolites like Acetyl-CoA, citrate, succinate, α-ketoglutarate, fumarate, and malate among others, that act as second messengers and regulate diverse aspects of the cell function [8].

A decline in mitochondrial quality and activity is associated with normal aging and correlates with development of a wide range of age-related diseases [9]. Mitochondrial functions respond and adapt to different circumstances, stages of development, proliferation, tissue-specific functions, and temperature changes. They are coordinated by the nucleus through signaling mechanisms in order to cover the energetic needs of the cells, this communication is called anterograde regulation. Mitochondria can also communicate with the nucleus and modulate genetic expression of nuclear encoded genes. This phenomenon is known as the retrograde response. Through this mito-nuclear communication cellular homeostasis is ensured, different stress responses are activated and organismal functions and lifespan are maintained [10]. 

Most mitochondrial proteins are synthesized as precursors in the cytosol and subsequently imported through the mitochondrial import machinery; the translocase of the outer membrane, TOM complex, and the translocase of the inner membrane, TIM complex. Because some subunits of the electron transport chain (ETC) are encoded in the mitochondrial DNA (mtDNA) [11], imbalances between subunits of the OXPHOS machinery that need to be assembled in membrane complexes, generate an accumulation of hydrophobic proteins within the mitochondria causing proteotoxic stress. The mitochondrial protein import machinery plays an important role as regulatory hub under normal and stress conditions [11]. In addition, lipid composition of mitochondrial membranes is essential for the good functioning of the organelles. The majority of lipids are synthesized in ER and extensive exchange of lipids and their precursors occurs between the ER and mitochondria as well as between mitochondrial membranes [12]. 

Mitochondrial prohibitins, PHB-1 and PHB-2, belong to the SPFH (stomatin/prohibitin/flotillin/HflKC) family of proteins [13]. SPFH-family members present across prokaryotic and eukaryotic life [14], are membrane-anchored and perform diverse cellular functions in different organelles. Within mitochondria, the PHB complex has been associated to mtDNA maintenance, protein synthesis and degradation, assembly of the OXPHOS system, maintenance of cristae structure, and apoptosis [15]. This diversity of phenotypes associated to PHB depletion could reflect different consequences of losing one unique biochemical function that remains to be fully clarified. Prohibitin deficiency shortens lifespan in wild-type *Caenorhabditis elegans* but dramatically extends lifespan in a variety of metabolically compromised animals [16], linking prohibitin functions in mitochondria with cellular metabolism [17]. Lack of PHB complex induces a strong mitochondrial stress response, the so-called mitochondrial unfolded protein response (UPR^mt^) [18,19,20,21], through a non-canonical mechanism [21]. Interestingly, under conditions where lifespan is drastically increased upon PHB depletion, the PHB-mediated induction of the UPR^mt^ is suppressed [20], suggesting that metabolic stress confers protection against PHB depletion and different signaling mechanisms might be at play.

Our knowledge of the different retrograde signaling pathways that evolved in different organisms and in response to different insults is continuously increasing, which will help understanding the complex relation between mitochondrial function, stress responses and longevity. Herein, we will review the main mitochondrial stress response pathways and their impact on aging, with particular focus on the UPR^mt^ and the mechanisms described in *C. elegans*. In the last section, we review the literature that points to PHBs as important players of mitochondrial quality control systems. PHB proteins respond to mitochondrial stress, functioning in the stabilization of mitochondrial membrane proteins, in membrane biology, and in mitochondrial degradation.

Understanding how mitochondria responds to internal and external stimuli is of importance to understand how organisms respond to stress and the mechanisms behind the role of mitochondria in health, aging and disease. 

## 2. Mitochondrial Stress Responses

Proper mitochondrial activity is preserved through regulation of mitochondrial dynamics, fusion and fission, modulation of mitophagy and maintenance of mitochondrial homeostasis through activation of the mitochondrial unfolded protein response, UPR^mt^. 

### 2.1. Mitochondrial Dynamics 

Mitochondrial dynamics modulate number, morphology and functions of the mitochondria in order to adapt to the actual energy demand and to the availability of resources. Under conditions of high energy demand and low energy supply, such as starvation, mitochondrial fusion is very active, mitochondria show elongated morphology and ATP production is more efficient. On the contrary, in situations of low energy demand and high energy supply, mitochondrial fission is activated leading to fragmented mitochondria with decreased ATP production and elevated ROS production [22]. 

Mitochondrial dynamics are important for stress responses as damaged mitochondria can fuse for the exchange of material and mitochondrial fission allows segregation of damaged mitochondria and is necessary for mitophagy. 

In *C. elegans*, mitochondrial fission requires the dynamin-related protein DRP-1 that controls the scission of the mitochondrial outer membrane [23] and mitochondrial fusion requires the mitofusin FZO-1 for outer membrane fusion [24] and the homologue of OPA1, EAT-3 for inner membrane fusion [25]. Importantly, mutations in genes involved in mitochondrial dynamics result in altered mitochondrial function [26]. Nematodes with mutations in *drp-1* or *fzo-1* show reduced maximal respiratory capacity at the fourth larval stage. In addition, mitochondrial dynamics related genes, even though they are not major drivers of longevity, they are important for the enhanced lifespan of insulin mutants [27,28]. In *C. elegans* the only insulin receptor is encoded by *daf-2* and mutants for this gene have altered metabolism and are long-lived. Defects in mitochondrial fusion, by depleting *fzo-1* or *eat-3*, reduce lifespan of *daf-2(e1370)* mutants [27] while defects in mitochondrial fission, by depleting *drp-1*, further increase *daf-2* lifespan [28]. Furthermore, depletion of *drp-1* affects lifespan of mitochondrial defective mutants [28]. Mutations in the iron sulfur protein (ISP-1) of mitochondrial complex III, *isp-1(qm150)*, and mutations CLK-1, an enzyme required for ubiquinone biosynthesis and thus for respiration, *clk-1(e2519)*, reduce oxygen consumption [26,29] and enhance lifespan, which is further enhanced by defects in mitochondrial fission [28]. These results suggest that mitochondrial dynamics play an essential role in maintaining the enhanced longevity of certain mutants. This role might be conserved through evolution as components of the mitochondrial fusion/fission machinery are highly conserved [28].

### 2.2. Mitophagy

To counteract the accumulation of defective mitochondria, cells activate mitophagy, a mechanism by which defective mitochondria are selectively detected and degraded [30]. Mitophagy can be ubiquitin dependent or ubiquitin independent [31]. The majority of studies revealing molecular mechanisms involved in mitophagy have been performed in yeast and mammalian cells. However, it is known that in *C. elegans* the ubiquitin dependent pathway is regulated by the phosphatase and tensin homologue (PTEN)-induced putative kinase 1, PINK-1, and the cytosolic E3 ubiquitin ligase homologue of human parkin (PARK2), PDR-1 [32]. In addition, the mitophagy receptor DCT-1, a putative orthologue of the mammalian NIX/BNIP3, has been shown to act in the same pathway as PINK-1 and PDR-1 to regulate mitophagy in nematodes. DCT-1 is ubiquitinated by PDR-1 and its ubiquitination is dependent on PINK-1 [32].

Under basal conditions, PINK-1 is imported through the translocase of the outer membrane, TOM, and the translocase of the inner membrane, TIM, where it is cleaved and subsequently degraded. On the contrary, when mitochondria are depolarized, PINK-1 cannot be imported to the inner mitochondrial membrane (IMM) and it is stabilized in the outer mitochondrial membrane (OMM) [33]. The accumulation and auto-phosphorylation of PINK-1 facilitates the recruitment of Parkin/PDR-1 to the mitochondria and induces its E3 ubiquitin ligase activity. Once active, PDR-1 ubiquitinates OMM proteins, recruiting autophagy adapter proteins to mitochondria and targeting mitochondria for selective autophagy [34]. 

Mitophagy is a conserved mechanism to prevent transmission of paternal mtDNA to the progeny, therefore preventing heteroplasmy [35,36,37]. This mechanism is conserved through evolution and appears to be necessary as mice with high heteroplasmy have altered metabolism and reduced cognitive functions [38]. However, the mechanism through which paternal mitochondria are degraded remains still elusive [39]. While mitochondria from worm sperm are not ubiquitinated [36], ubiquitin chains play a role as degradative signal in flies [40]. The mitochondrial endonuclease G is important for elimination of paternal mitochondria both, in worms and flies [41,42]. However, different mechanisms, species specific, are involved in the transmission of only maternal mtDNA and further investigation is needed. 

Mitophagy has been described as a common longevity assurance process as depletion of mitophagy components, while it does not affect lifespan of wild type animals, it shortens the enhanced lifespan of mutants of the insulin receptor DAF-2, *daf-2(e1370)*, as well as the increased lifespan of mitochondrial defective mutants, *isp-1(qm150)* and *clk-1(e2519)* [32]. Disruption of mitophagy leads to pronounced mitochondrial defects, such as distorted mitochondrial network, mitochondrial membrane depolarization, reduced ATP levels, elevated oxygen consumption and elevated ROS levels [32]. Interestingly, DCT-1 expression is regulated by two transcription factors, DAF-16 and SKN-1. Compared to wild type, *daf-2(e1370)* mutants have increased levels of mitophagy which are reduced in the absence of SKN-1 [32]. In addition, mitophagy is protective against *Pseudomonas aeruginosa* infection because depletion of PINK-1 increased lethality upon *Pseudomonas* infection [43]. In addition, mitochondrial dysfunction confers protection from rotenone-induced neurodegeneration in a cell-non-autonomous manner via p38MAPK/ATF-7. The authors proposed that p38MAPK-mediated immunity activates mitophagy to confer neuroprotection [44]. Moreover, mitophagy is required for lifespan extension under iron starvation [45] showing again the protective role of mitophagy. 

### 2.3. The Mitochondrial Unfolded Protein Response, UPR^mt^

Even though originally discovered in mammalian cells, the mitochondrial unfolded protein response signaling pathway has been extensively studied in the nematode *C. elegans*. For an overview of the mammalian UPR^mt^ we direct the readers to a recent review [46]. In this work, we will review in detail the components of the mitochondrial stress response described in *C. elegans* (Figure 1).

Under stress conditions like depletion of mtDNA [47], loss of mitochondrial membrane potential, imbalance between nuclear- and mitochondrial-encoded proteins [48] or accumulation of unfolded proteins within the mitochondria, there is activation of a retrograde signaling whereby the nucleus is informed to induce a transcriptional program aimed at restoring mitochondrial function. This response is known as the unfolded protein response of the mitochondria, UPR^mt^, and it is activated to maintain mitochondrial proteostasis by inducing the expression of mitochondrial chaperones and proteases that control protein folding, assembly and degradation.

Accumulated unfolded and/or unassembled proteins in the mitochondrial matrix are cleaved by the mitochondrial protease, ClpP [49] and the resulting peptides are exported to the cytoplasm through the mitochondrial ATP-binding cassette (ABC) transporter, HAF-1 [50]. The accumulation of peptides in the cytosol, in combination with a reduced efficiency of mitochondrial import in general and in particular import of the transcription factor ATFS-1, possibly modulated by HAF-1, lead to the nuclear translocation of s ATFS-1 [51]. Once in the nucleus, ATFS-1 together with DVE-1 and UBL-5 [49,52], induce the expression of genes involved in restoring mitochondrial homeostasis such as mitochondrial chaperones (*hsp-6* and *hsp-60*), proteases (*clpp-1*, *lonp-1*, *spg-7* and *ymel-1*), the fission factor *drp-1* and mitochondrial transporters (*tim-23* and *tim-17*) [51,53]. 

Moreover, ATFS-1 induces the expression of glycolysis related genes, promoting an alternative form of ATP production, and negatively regulates the expression of multiple TCA cycle and OXPHOS genes. Interestingly, in addition to the nuclear localization signal (NLS), ATFS-1 presents a mitochondrial targeting sequence (MTS) and is normally imported to the mitochondria where it is degraded by the Lon protease. During stress, even though the majority of ATFS-1 is translocated to the nucleus due to defective mitochondrial import, a percentage of ATFS-1 also accumulates inside the mitochondria and binds mtDNA where it limits mitochondria-encoded mRNA accumulation [53]. Thus, in addition to promote mitochondrial protein homeostasis, the transcription factor ATFS-1 has been proposed to act as a metabolic regulator and to assist in the complete recovery from mitochondrial dysfunction. 

Furthermore, studies from different labs have shown protective roles for mitophagy and the UPR^mt^ during bacterial infection [53,54,55,56,57]. Exposure to *Pseudomonas aeruginosa* induces an innate immune response similar to the response induced by mitochondrial dysfunction, i.e. induction of transcription of mitochondrial chaperones and secreted lysozymes. ATFS-1 appears as a key regulator of the protective innate immunity and worms with hyperactive ATFS-1 have better clearance of *Pseudomonas* from the intestine and enhanced survival [55,56]. 

Besides changes in the genetic expression, mitochondrial stress causes wide-spread changes in chromatin structure [58,59]. Mitochondrial stress causes global condensation of the chromatin as shown by a reduced size of nuclei upon depletion of the nuclear-encoded cytochrome *c* oxidase subunit CCO-1 [59]. The histone methyltransferase MET-2 and the nuclear co-factor LIN-65, that together trigger the di-methylation of the histone H3K9, are required for the induction of the UPR^mt^. Although those changes globally cause genetic repression, they are needed for DVE-1 to translocate to the nucleus and to bind to the opened free regions [59]. It could be that DVE-1 is passively forced to bind loose regions of the chromatin or could be that other chromatin modifier factors are involved in the mechanism. By RNA-seq Tian et al. demonstrated that the majority of genetic changes elicited by *cco-1(RNAi)* required *met-2* and *lin-65* [59]. 

In addition, two conserved demethylases, the jumonji family proteins JMJD-1.2 and JMJD-3.1, have been shown necessary for the induction of the UPR^mt^ upon depletion of *cco-1*, probably by removing the repressive H3K27 methylation marks from coding regions in UPR^mt^-related genes [58]. Interestingly, overexpression of the two demethylases is sufficient to induce the UPR^mt^, albeit core components of the UPR^mt^ are required [58]. Importantly, *cco-1(RNAi)* induces the expression of *jmjd-1.2* and *jmjd-3.1*, placing them downstream of mitochondrial defects [58]. Further studies are needed to understand the complex relation between mitochondrial dysfunction and epigenetic changes as JMJD3 and PHF8, the mammalian homologues of JMJD-3.1 and JMJD-1.2 respectively, belong to the family of 2-oxoglutarate dependent oxygenases [60] and could be induced by high levels of intermediates of TCA cycle.

Strikingly, the mitochondrial stress response can be activated in a cell non-autonomous manner. Perturbing mitochondrial function in the nervous system induces the UPR^mt^ in peripheral tissues [61,62,63]. The induction of the mitochondrial stress response caused by neuronal mitochondrial dysfunction needs the UPR^mt^ components both in neurons and periphery cells. Additionally, by performing a large scale CRISPR-Cas9 screen targeting 103 neuropeptide genes, Shao et al. described FLP-2 as a possible mediator of this non-autonomous mechanism from the sensory neurons ASK, AWA and AWC, to the peripheral tissues, passing through the interneuron AIA [62]. Moreover, Berendzen et al. reported dense core vesicle secretion to be essential for the induction of the UPR^mt^ in peripheral tissues in a Huntington disease model (PolyQ40). More specifically, they identified serotonin as the only amine to be required for the communication between the affected neurons and the peripheral tissues [63]. A recent study has involved retromer dependent Wnt signaling in the cell non-autonomous response [64]. The authors described that VPS-35, a component of the retromer complex, is required for the polyQ40 non-autonomous induction of mitochondrial stress. Retromer complex regulates, in addition, the retrieval of cargo for efficient recycling of signaling receptors to allow proper transcellular signal transduction. They identified the Wnt secretion factor MIG-14 and the Wnt ligand EGL-20 to be necessary for the peripheral UPR^mt^. Furthermore, neuronal expression of EGL-20 is sufficient to induce UPR^mt^ in peripheral tissues even in the absence of mitochondrial stress. Interestingly, serotonin is required for this induction. Nevertheless, serotonin is not sufficient for the cell non-autonomous induction of the UPR^mt^ and not all mitochondrial perturbations induce the cell non-autonomous response, thus further investigation is needed to define other possible secreted factors and the exact molecular mechanism(s).

A recent study identified the sphingosine kinase SPHK-1 as an early indicator of the UPR^mt^ activation [65]. Previously, Liu et al. identified that proper synthesis of sphingolipid was required for activation of UPR^mt^, as knock down of *sptl-1* suppressed the response elicited upon treatment with antimycin A [56]. More in particular, they reported that animals defective in ceramide biosynthesis are deficient in mitochondrial surveillance [56]. Being ceramide a precursor of sphingosine (SPH), Kim and Sieburth assessed the role of SPH and its derivate sphingosine 1 phosphate (S1P) in the mitochondrial stress response [65]. In this study, they showed that SPHK-1 associates with mitochondria upon mitochondrial stress and transforms SPH in S1P. The production of S1P rapidly activates the UPR^mt^ even if mitochondrial insults are transient. In addition, expression of SPHK-1 in the intestine is required for the cell non-autonomous activation of the UPR^mt^. 

### 2.4. Cytosolic Responses Reacting to Mitochondrial Proteotoxic Stress

In addition to the UPR^mt^, which acts mainly at a transcriptional level, recent studies have identified different cytosolic responses reacting to mitochondrial proteotoxic stress. The mitochondrial precursor over-accumulation stress (mPOS) [66] and the UPR activated by mistargeting of proteins (UPR^am^) [67] result in a simultaneous inhibition of cytosolic protein synthesis and an increase of proteasomal activity. The two responses coordinate the inhibition of cap-dependent translation by down-regulating ribosomal proteins, the mTOR pathway and mRNA turnover. These responses have only been described in the yeast *Saccharomyces cerevisiae* and further investigation is required to decipher the exact mechanism. However, in worms an attenuation of protein synthesis has been demonstrated. By performing an RNAi screen focusing on kinases and phosphatases, Baker et al. identified GCN-2 as a component that when knocked down further increases the UPR^mt^ of *clk-1* mutants. Under mitochondrial stress, the kinase GCN-2 phosphorylates the translation initiation factor eIF2α in order to reduce protein translation, which is crucial for maintenance of mitochondrial functions [29]. Interestingly, this cytosolic response acts in parallel to ATFS-1, as depletion of *atfs-1* does not modulate phosphorylation levels of eIF2α under mitochondrial stress. These results support a role for translation attenuation in promoting mitochondrial protein homeostasis, as it has been previously shown for endoplasmic reticulum homeostasis [68].

The mitochondrial to cytosol stress response (MCSR) is an additional mechanism to restore proteostasis and to reduce the proteotoxic effect in worms [69]. Kim et al. observed that mitochondrial disruption by depleting *hsp-6* expression, in addition of causing mitochondrial stress, it induces cytosolic chaperone expression. This genetic manipulation triggers a restructuring of fat metabolism, resulting in accumulation of lipids and cardiolipin and inhibition of ceramide synthesis. Interestingly, treating worms with perhexiline (PHX), an inhibitor of the carnitine palmitoyltransferase, which blocks fatty acid oxidation and provokes accumulation of fatty acid, induces the MCSR. On the contrary, high levels of ceramide blocks the MCSR. Thus, the shift in fat metabolism facilitates crosstalk between mitochondria and cytosol in order to improve cytosolic protein homeostasis, as it slows the progression of motility defects in polyQ-expressing animals. Interestingly, the MCSR needs the cooperation of DVE-1 and HSF-1 to induce the cytosolic response upon mitochondrial perturbations [69].

A recent study revealed a new interorganellar mechanism for proteostasis regulation in response to stress in yeast, which the authors called interorganellar proteostasis transcription program (IPTP) [70]. They constructed variants of the mitochondrial ribosomal protein Mrps12 and described one with hypoaccurate mitochondrial translation and one with hyperaccurate mitochondrial translation. They revealed that mitochondrial translation accuracy impacts the proteostatic capacity of the cytosol and affects stress responses, having consequences in aging. Hypoaccurate translation impairs cytoplasmic quality control systems and shortens lifespan, possibly due to a collapse of the system. In contrast, hyperaccurate mitochondrial translation increases the capacity of the cytoplasmatic proteostasis system and leads to increased lifespan. This study provides a new demonstration of the interorganellar communication in order to maintain proteostasis. However, more research is need to identify the molecular signal(s) from mitochondria to the rest of the cell.

## 3. Mitochondrial Unfolded Protein Response and Its Impact on Aging

Mitochondrial function is key for organisms’ survival, playing a central role in metabolic homeostasis, however the link between mitochondrial dysfunction and aging is more complicated than initially thought. Although many mitochondrial perturbations that induce the UPR^mt^ have been shown to extend lifespan in yeast, worms, flies and mice [71,72,73,74], the exact link between UPR^mt^ activation and longevity remains unclear and controversial [19].

In *C. elegans* two studies performing RNAi screens reported that animals with mitochondrial dysfunction are long-lived [72,75]. Depletion of genes involved in respiration and ATP production, such as *atp-3*, *nuo-2*, *cyc-1*, *cco-1*, *clk-1*, and *isp-1*, led to reduced growth rate and body size, slowed behavioral rates and enhanced lifespan [72,75]. Importantly, lack of these genes extends lifespan in an insulin independent manner. 

Interestingly, depletion of mitochondrial components only during development is sufficient to enhance lifespan, whereas reduction of gene expression during adulthood, even though it decreases ATP production, it does not affect lifespan [48,61,75,76]. In addition, it is worth noting that some disruption of mitochondrial function specifically in the neurons, not only induced the UPR^mt^ in the intestine, but was also able to increase lifespan [61,62]. However, not all mitochondrial perturbation in the nervous systems affects lifespan in the same manner [62,63,64], so further investigation is needed in order to study how different disruptions of mitochondrial function affect metabolic and aging rates. 

Intriguingly, proper expression of core components of the mitochondrial stress response, such as UBL-5, HAF-1, DVE-1, ATFS-1, JMJD-1.2, JMJD-3.1, MET-2, LIN-65, and GCN-2, are needed for the enhanced lifespan of mitochondrial defective mutants [29,48,51,58,59,61]. Interestingly, JMJD-1.2 and JMJD-3.1 are required not only during development but throughout the entire life for the UPR^mt^ mediated enhancement of lifespan [58]. Indeed, over-expression of the two demethylases is sufficient to prolong lifespan [58]. In the same direction, lifespan enhancement upon mitochondrial stress requires *lin-65* and *met-2* [59]. However, chromatin reorganization appeared to act synergistically with induction of the UPR^mt^ to regulate lifespan as only by combining loss of *met-2* with *atfs-1(RNAi)* the enhanced lifespan of *cco-1* depleted worms is reduced to wild type levels [59]. This suggests that mitochondrial perturbation early in life establishes an epigenetic memory that ensures the protection from future insults and maintains the beneficial effects throughout lifespan [58]. 

Nevertheless, while a number of mitochondrial perturbations induce the UPR^mt^ and increase lifespan, there are also evidences that the UPR^mt^ may not be sufficient by itself to extend lifespan [19]. After performing an RNAi screen looking for inducers of the mitochondrial stress response, they found that depletion of half of the candidates extends lifespan while depletion of about 30% of the candidates reduces it. Among the last candidates, they described mostly genes involved in mitochondrial import, such as TOMM-22, DNJ-21, TIN-44, TIMM-17B.1, and TIMM-23 [19]. In addition, authors show that a gain-of-function mutation in *atfs-1* does not extend lifespan [19].

Through RNAi dilution experiments, Rea et al. described that mild knockdown of ETC genes extends lifespan while too strong knockdown reduces lifespan [76]. Along this line, the lack of correlation of UPR^mt^ induction and lifespan could be at least partially explained by the mitochondrial threshold effect [77], with strong induction reflecting irreparable damage and short lifespan, while milder induction could trigger cellular defense mechanisms resulting in lifespan extension.

Mitochondria are also a major source of ROS, what have been determined as the primary cause of damage in macromolecules such as DNA, proteins and lipids. The mitochondrial free radical theory of aging states that aging is the result of accumulation of oxidative damage caused by free radicals generated as by-products of normal metabolism, mainly from mitochondria [78]. Nevertheless, this theory is in continuous re-evaluation during the last decade due to new insights that suggest a more complex relation between free radicals and aging [79]. Knockout of the mitochondrial superoxide dismutase SOD-2 in *C. elegans* increases oxidative stress and prolongs lifespan [80]. The new proposal is that ROS generation, instead of being the initial prompt of the aging process, represents a stress signal in response to age-dependent damage, initiating different molecular mechanisms in order to alleviate damage. However, ROS accumulation reaches a level at which it becomes toxic and starts to contribute to damage [79].

It seems that when the mitochondrial stress is too high, the protective effects of UPR^mt^ are insufficient to counteract the damage, thus the beneficial adaptive response becomes maladaptive. 

## 4. Mitochondrial Prohibitins, Key Players in Mitochondrial Quality Control Mechanisms

Prohibitins are strongly evolutionarily conserved mitochondrial proteins whose true biochemical function still remains unknown [18,81]. Its high degree of conservation suggests an important cellular function; however, prohibitin deletion does not cause any observable growth phenotype in the unicellular yeast *S. cerevisiae* [82]. By contrast, prohibitins are required for embryonic development in *C. elegans* [83] and in mice [84], while their postembryonic depletion by RNAi in *C. elegans* causes severe germline defects [83] suggesting an essential role in cell proliferation. The prohibitin (PHB) family is composed of two subunits PHB1 (32 KDa) and PHB2 (34 KDa), that physically associate with each other to form a large multimeric complex of approximately 1 MDa [85]. PHB2 has a N-terminal trans-membrane domain and the N-terminal of PHB1 is expected to be membrane associated. Both proteins contain the conserved PHB domain, common to other scaffold proteins such as stomatin and flotillin, and a coil-coiled C-terminal domain responsible for the interaction between PHB1 and PHB2 [86]. Approximately 14 heterodimers become associated and form a ring-like structure in the IMM, projected into the mitochondrial intermembrane space [85]. Loss of either of the subunits leads to the absence of the whole complex, both in unicellular and multicellular eukaryotes [83,87]. 

The PHB complex has been assigned a variety of putative functions within mitochondria. In the sections below, we summarize all genetic and physical interactions described for the PHB complex that support a prominent role for this complex in mitochondrial quality control, participating in mitochondrial biogenesis and degradation (Figure 2) and responding to mitochondrial stress.

### 4.1. The PHB Complex in Mitochondrial Turnover

Several lines of research point to a critical role for PHB proteins in mitochondrial biogenesis. Both PHB subunits have been involved in the organization and stability of mitochondrial nucleoids together with mtDNA-binding proteins, TFAM, and mtSSB, as well as metabolic enzymes [88,89,90]. In addition, PHB proteins have been found interacting with ATAD3 in human cells [91], another mtDNA interacting protein. The authors showed that depletion of either PHB or ATAD3 results in dramatically reduced mitochondrial protein synthesis, supporting the notion that mitochondrial nucleoids and mitochondrial transcription are linked to translation. ATAD3 has been shown to control cristae structure, influencing mtDNA replication and cholesterol levels, connecting mitochondria with the endoplasmic reticulum (ER) in cholesterol transport [92,93,94].

In yeast, prohibitins were shown to stabilize newly synthesized mitochondrial translation products of the ETC [95]. Direct binding to complex IV subunits of the ETC was observed, and a role in protein complex assembly, acting as a holdase-unfoldase type of chaperone, was proposed [95]. Later, PHB2 was shown to interact with sphingosine-1-phosphate to regulate complex IV assembly in mice mitochondria [96]. Also, in mammals, the PHB complex has been suggested as assistant for incompletely assembled complex I subunits of the ETC [97]. A genetic and physical interaction of prohibitins with mitochondrial m-AAA proteases in yeast, suggests a role for the PHB complex in protecting newly imported proteins from degradation by m-AAA protease [98]. Moreover, a genetic interaction with Atp23, a processing peptidase and chaperone for the F_1_F_O_-ATP Synthase, in yeast, further supports a role for the PHB complex in ETC biogenesis [99]. In addition, PHB proteins are also important for the formation of respiratory super-complexes together with stomatin-like protein 2 (SLP2) in mammalian cells [100,101]. In a proteomic analysis in mouse embryonic fibroblasts (MEF), PHB2 was also found to interact with SLP2 [102]. In this same study, additional interactions of PHB2 where detected, including several proteins of the mitochondrial inner membrane involved in the import of nuclear-encoded proteins such as the translocase subunits TIM22 and TIM23, and components of the Pam import motor, Pam16p and DNAJC19/DNJ-21/Pam18p, that helps mtHSP70 to pull polypeptides into the mitochondrial matrix. Interaction with the AAA proteases SPG7, AFG3L1/2 and YME1L was also detected, as well as with OXPHOS and ATP synthase subunits [102].

In addition to their relevant role in mtDNA maintenance, mitochondrial protein synthesis, degradation and assembly, strong genetic interactions of prohibitins with genes modulating mitochondrial phospholipid biosynthesis, in particular cardiolipin (CL) and phosphatidylethanolamine (PE) have been observed in yeast [82,103,104], suggesting a role for the PHB complex as a membrane organizer affecting the distribution of CL and PE by clustering them at distinct sites of the IMM. Further substantiating the functional link between PHB complexes and mitochondrial membrane lipids, lack of PHB complexes altered cardiolipin acylation in MEFs, while the transcriptional response of PHB2-deficient cells showed altered lipid metabolism, most prominently cholesterol biosynthesis [102]. This alterations in CL, PE and cholesterol may affect membrane protein and mtDNA stability. Prohibitins are required for the formation of mitochondrial cristae, by stabilizing long forms of the dynamin-like GTPase OPA1, an essential component of the mitochondrial fusion machinery [84], which could explain the fragmentation of the mitochondrial network that occurs upon depletion of PHB proteins in nematodes and MEFs [83,84]. Loss of PHB complexes in mice neuronal cells results in aberrant cristae morphogenesis that cannot be rescued by stabilization of long OPA1 isoforms, suggesting a more direct role of the PHB complex in keeping the structure of the IMM.

Intracellular organelles maintain their homeostasis through the continuous exchange of lipids and Ca^2+^. In yeast, the ERMES (ER mitochondria encountered structures) complex is involved in the transport of phospholipids and Ca^2+^ between the ER and mitochondria [105]. Another mitochondrial complex named MINOS, MitOS, or MICOS has been identified associated to the IMM and it has been proposed to form the central core of a large organizing system named ERMIONE, which includes the ERMES complex, PHB ring-like structures, the TOM and TIM translocases, and Mdm31/32 proteins required for mtDNA maintenance [106]. PHB proteins have been genetically linked to MICOS and ERMES complexes in synthetic lethal screens in yeast [82,103,105]. Therefore, PHB might belong to the ER-mitochondria organizing network linking the ER and the two mitochondrial membranes to maintain membrane architecture, mtDNA, as well as ion, protein and lipid homeostasis, explaining the variety of phenotypes observed upon PHB depletion in different model systems. Apart from the ER, mitochondria are in constant exchange with other organelles, including peroxisomes, lysosomes and lipid droplets [3,4]. Interestingly, PHB subunits have been found as transient interactors of the peroxisomal importomer [107], as well as in lipid droplets and lipid rafts preparations [108,109].

Regarding mitochondrial degradation, PHB proteins have been found in a variety of systems to be poly-ubiquitinated during spermiogenesis, suggesting a possible role in elimination of paternal mitochondria [110,111,112]. Recently, PHB2 has been reported as a mitochondrial receptor essential for Parkin mediated mitophagy, in mammalian cells. PHB2 contains a LC3-interacting domain. In response to a mitochondrial uncoupling agent and following proteasome-mediated degradation of the OMM, binding of LC3 to PHB2 would ensure mitochondrial clearance by recruiting the autophagic machinery to the IMM [37]. Proximity of LC3 to PHB2 has also been reported in PINK1 and Parkin independent mitophagy [113].

### 4.2. The PHB Complex Responds to Mitochondrial Stress

PHB protein levels respond to different mitochondrial stress conditions suggesting an important role for the PHB complex in mitochondrial homeostasis. In mammals and in *C. elegans*, the level of PHB proteins increase in response to an imbalance in the synthesis of mitochondrial- and nuclear-encoded mitochondrial proteins as a result of inhibition of mitochondrial translation [83,114]. Depletion of assembly factors of the ETC cause proteotoxic stress by the accumulation of unassembled subunits. In yeast, protein levels of the PHB complex increase in mutants with compromised biogenesis of cytochrome c oxidase. Mss51p is a COX1 mRNA-specific processing factor, translational activator and chaperone [115], which deletion results in increased PHB complex [95]. Similarly, in yeast cells mutant for Shy1p/SURF1, a complex IV assembly factor causing Leigh syndrome in humans when mutated, the PHB complex is overexpressed [116]. All these observations point to a critical role for the PHB complex in situations of mitochondrial proteotoxicity, plausibly to hold and protect mitochondrial membranes from unassembled hydrophobic ETC subunits. 

In addition, high level expression of both proteins is consistently seen in primary human tumors, and thus, responding to situations of altered metabolism and/or high proliferation [114]. Prohibitins have been shown to accumulate in response to several stresses, including chemotherapeutic agents, the UPR^ER^ inducer tunicamycin and nutrient starvation, while PHB knock-down sensitized melanoma cells [117]. Several reports suggest PHB proteins as potential druggable targets to halt cancer proliferation and metastasis, while small molecules exhibiting antitumor effects have been shown to target prohibitins [118,119].

### 4.3. The PHB Complex and Lifespan

PHBs have been related to several age-related diseases like cancer, neurodegenerative and metabolic disorders [117,118,120,121,122,123]. Loss of PHB decreases resistance to apoptosis [84] and accelerates aging in yeast [124] and mammalian cells [125,126,127]. In mice, treatment with the NAD(+) precursor nicotinamide riboside (NR) extends lifespan and enhances the self-renewal capacity of muscle stem cells through induction of PHB proteins and the UPR^mt^ [128]. 

Strikingly, PHB depletion shows opposite aging phenotypes across phyla. Originally described in *C. elegans*, lack of PHB decreases the lifespan of wild type animals, whereas it extends lifespan in a variety of compromised conditions such as in mitochondrial mutants, in fat metabolism mutants, in dietary restricted mutants and in mutants defective in either of the two diapause signaling pathways, TGF-β signaling and insulin/IGF-1 signaling [16,20]. Similarly, PHB depletion causes lifespan extension in *S. cerevisiae* in response to caloric restriction [129]. 

Lack of PHB induces a very strong UPR^mt^ in *C. elegans* [18,19,20,21]. Interestingly, PHB depletion induces the UPR^mt^ in a background dependent manner. While the UPR^mt^ induction upon PHB depletion in otherwise wild type animals is very strong, in the insulin mutant *daf-2(e1370)* and the mTORC2 mutant *sgk-1(ok538)*, where lifespan is drastically increased upon PHB depletion, the PHB-mediated induction or the UPR^mt^ is suppressed [20]. This is reminiscent of the threshold effect observed upon mitochondrial dysfunction and suggests that conditions in which PHB depletion extends lifespan might protect against mitochondrial stress.

## 5. Future Prospects

Prohibitin deficiency signals to the nucleus by a non-canonical UPR^mt^ pathway, as HAF-1 and UBL-5 are not required. Instead, depletion of HAF-1 or UBL-5 further induces the UPR^mt^ in PHB mutant nematodes [21]. Interestingly, PHB depletion results in opposing aging phenotypes depending on the metabolic status of the animals [16], accompanied by a differential regulation of the UPR^mt^ in the cases studied [20]. Identifying which mitochondrial-to-nucleus signaling mechanisms are involved in the different conditions will help understanding the opposite longevity phenotypes caused by PHB depletion. Depletion of proteins involved in the transport of nuclear-encoded mitochondrial proteins reduces lifespan and strongly induces the UPR^mt^ [19]. These included *tomm-22*, a component of the outer membrane translocase TOM, and several components of the TIM and PAM complexes that function to transport proteins into or across the inner membrane with the help of the *hsp-6*/mtHsp70 chaperone. Among them, F15D3.7/TIM23, F45G2.8/TIM16/PAM16, and DNJ-21/DNAJC19/PAM18, have been shown to physically interact with the PHB complex [102]. It would be interesting to study if the same opposite lifespan phenotype can be seen upon depletion of mitochondrial components of the import machinery. 

A role for the PHB complex in mtDNA stability, OXPHOS biogenesis and assembly, mitochondrial cristae architecture, metabolism and membrane lipid homeostasis has been demonstrated. Whether the PHB complexes regulate all these processes acting as holdase/unfoldase type of chaperones, protein/lipid scaffolds, or a more direct role in in keeping mitochondrial cristae junctions, remains to be deciphered. Advances in structural biology of membrane proteins hold promise in providing insights into the molecular architecture and the biochemical role of the PHB complex [130].

While the exact biochemical function of this complex remains to be elucidated, it is clear that different phenotypes can be observed upon PHB depletion depending on cell types and the metabolic status of the organisms affected. As a consequence, different mitochondrial retrograde signaling could be triggered in response to stress. A deeper elucidation of how mitochondria communicate with the nucleus upon different stresses is crucial to understand how the cell responds and adapts to keep homeostasis. Moreover, understanding the relationship between mitochondrial stress responses and lifespan is of fundamental importance to understand mitochondrial associated diseases and aging. 

## Figures and Tables

**Figure 1 cells-07-00238-f001:**
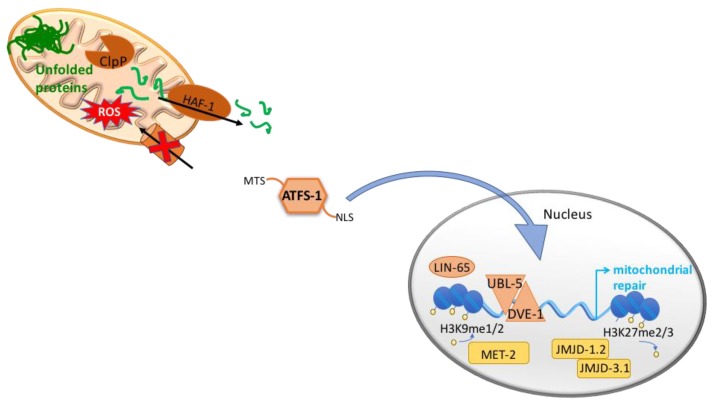
The mitochondrial unfolded protein responses, UPR^mt^. In order to maintain mitochondrial proteostasis, the UPR^mt^ is activated and induces the expression of mitochondrial chaperones and proteases. In *C. elegans*, unfolded proteins within the mitochondrial matrix are cleaved by the mitochondrial protease, ClpP, and the resulting peptides are exported to the cytoplasm through the mitochondrial ATP-binding cassette (ABC) transporter, HAF-1. ATFS-1 presents a mitochondrial targeting sequence (MTS) and a nuclear localization signal (NLS). Under normal conditions ATFS-1 is imported to the mitochondria where it is degraded by the Lon protease. However, during stress ATFS-1 is translocated to the nucleus due to defective mitochondrial import and to the accumulation of peptides in the cytosol. Once in the nucleus, ATFS-1 together with DVE-1 and UBL- 5, induce the expression of genes involved in restoring mitochondrial homeostasis. In addition, mitochondrial stress causes changes in chromatin structure. In particular, mitochondrial stress induces the histone methyltransferase MET-2 and the nuclear co-factor LIN-65 for the di-methylation of the histone H3K9. Moreover, two demethylases, JMJD-1.2 and JMJD-3.1, have been shown necessary for the induction of the UPR^mt^.

**Figure 2 cells-07-00238-f002:**
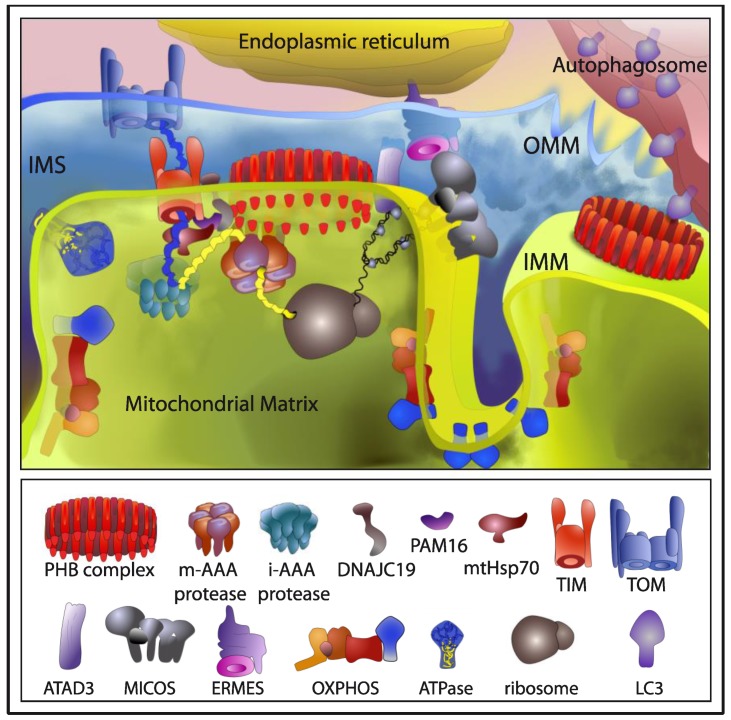
The role of the prohibitin (PHB) complex in mitochondrial homeostasis and turnover. Interactions involved in mitochondrial biogenesis are depicted in the left side, those involved in mitophagy in the right side. Both, physical and genetic interactions described for the PHB complex are depicted. PHB complex is involved in organization and stability of mitochondrial nucleoids together with ATAD3 in human cells. In yeast, as well as in mammals, prohibitins stabilize newly synthesized mitochondrial translation products of the electron transport chain (ETC), physically interacting with complex I and complex IV, acting as a holdase-unfoldase type of chaperone. In addition, PHB interacts with mitochondrial m-AAA proteases in yeast, suggesting a role for the PHB complex in protecting newly imported proteins from degradation by m-AAA protease. In a proteomic analysis, in mouse embryonic fibroblasts (MEF), PHB2 was found to interact with several proteins of the IMM involved in the import of nuclear-encoded proteins such as the translocase subunits TIM22 and TIM23, and components of the Pam import motor, Pam16p and DNAJC19/Pam18p, that helps mtHSP70 to pull polypeptides into the mitochondrial matrix. In synthetic lethal screens in yeast, PHB proteins have been genetically linked to MICOS and ERMES complexes. Therefore, PHB might belong to the ER-mitochondria organizing network linking the ER and the two mitochondrial membranes to maintain membrane architecture, mtDNA, as well as ion, protein and lipid homeostasis. Finally, binding of LC3 to PHB2 would ensure mitochondrial clearance by recruiting the autophagic machinery to the IMM following proteasome-mediated degradation of the OMM. TIM: translocase of the inner membrane; TOM: translocase of the outer membrane; OXPHOS: oxidative phosphorylation.

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
