# Peer review of "Mitochondrial Quality Control Mechanisms and the PHB (Prohibitin) Complex"

_cells, 2018, doi:10.3390/cells7120238_

Round 1

Reviewer 1 Report

This is a well-written, interesting review on mitochondria-to-nucleus signalling focusing on mitochondrial quality control in C. elegans. Current developments in the field are summarized and data from other model systems are incorporated. To further improve the manuscript, the authors could consider two main points:

1. The section on mitochondrial dynamics as presented now does not feel well integrated in the text and the main conclusions of the significance of mitochondrial dynamics for aging remain unclear. This section could benefit from further elaboration and a more vigorous interpretation.

2. A recent study  (Suhm T et al 2018) showed a tight connection between mitochondrial translation and cellular proteostasis, including a transcription program that apparently operates similarly to the mtUPR. This works should be discussed as it points to a highly conserved pathway.

Author Response

We would like to thank the Referee for the gratifying comments. 

Point 1:The section on mitochondrial dynamics as presented now does not feel well integrated in the text and the main conclusions of the significance of mitochondrial dynamics for aging remain unclear. This section could benefit from further elaboration and a more vigorous interpretation.

Response 1: We appreciate the Reviewer`s suggestion of expanding the mitochondrial dynamics section, which we have done now also including the role of mitochondrial dynamics in ageing. We hope the reviewer will now find the coverage of the mitochondrial dynamics section appropriate. 

Point 2:A recent study (Suhm T et al 2018) showed a tight connection between mitochondrial translation and cellular proteostasis, including a transcription program that apparently operates similarly to the mtUPR. This works should be discussed as it points to a highly conserved pathway.

Response 2: We very much appreciate the reviewer`s comment. We had missed that publication, which we are happy to have now included. A new paragraph has been incorporated in the section of “Cytosolic responses reacting to mitochondrial proteotoxic stress” and the reference has been added. We hope the reviewer finds our discussion satisfactory. 

Reviewer 2 Report

The review manuscript by Hernando-Rodríguez et al. thoroughly covers the signalling mechanisms behind different mitochondrial quality control pathways in C. elegans, with a focus on the role of mitochondrial quality control pathways in aging and on lifespan. The authors touch on a wide variety of molecular pathways and events that regulate mitochondrial health, including changes in mitochondrial dynamics, mitophagy, the UPRmt and cytosolic stress signalling pathways. A very detailed summary on the molecular functions of PHB complex proteins is provided and is accompanied by a targeted discussion on the potential roles of PHB complex function during mitochondrial stress responses and aging.

The authors’ comprehensive and clear explanations of a wide range of topics position this manuscript to be a very good introduction on the topic of mitochondrial quality control pathways and the role of the PHB complex in maintaining mitochondrial health. However, there are several grammatical errors that should be corrected prior to publication and several points that would benefit from further clarity in their description within section 2 and 3 of the manuscript.

Main comments:

1.     In section 2.2, the PINK-1 protein is occasionally referred to as PINK1 and the PDR-1 protein is sometimes referred to as Parkin. Consistent use of a single naming convention would help readers better follow along with the authors’ discussions. As the text is specifically discussing mitophagy in C. elegans, it would perhaps be beneficial to adopt the PINK-1/PDR-1 nomenclature that is unique to C. elegans.

2.     In the 4th paragraph of section 2.2, it is not clear from the text what the genetic background is of the “insulin mutants” and the “mitochondrial defective mutants” referred to by the authors. It is also not very clear what “hypoxia response” is referring to in this section.  It could also be helpful for readers if the authors provided some speculation on the relevance to mitophagy regulation and lifespan extension of the SKN-1/DCT-1 changes observed in the insulin mutant lines specifically mentioned in the text. 

3.     Section 2.3: The nuclear translocation of ATFS-1 in response to a disruption in its normal mitochondrial import has been well established as a central regulatory component of the UPRmt. Additionally, while there is not yet direct evidence showing the peptides released by HAF-1 interacting with mitochondrial import complexes, the actions of the HAF-1 transporter have been shown to cause a global repression in mitochondrial import which is critical for initiating the UPRmt through ATFS-1 (see PMID: 22700657 and PMID: 20188671 for both these points). While the authors do briefly mention that the nuclear import of ATFS-1 in response to disrupted mitochondrial protein import leads to the expression of UPRmt genes, and that peptides released by HAF-1 lead to ATFS-1 translocation to the nucleus, these points are mentioned in separate paragraphs and a connection between these two observations is not made clear in the text. As these series of events have been robustly shown to be central components of the UPRmt signalling upstream of ATFS-1 and DVE-1/UBL-5 mediated gene transcription, the authors’ description of UPRmt regulation in section 2.3 of the manuscript should be amended to reflect the importance of disrupted mitochondrial import of ATFS-1 to UPRmt induction.

Additionally, the disruption to mitochondrial import and subsequent change in ATFS-1 localisation are not referenced in Figure 1. Both the figure legend and the graphical components of Figure 1 should be updated to include descriptions of these events to more accurately represent the literature available on UPRmt signalling.  

Minor comments:

1.     Line 52: The sentence “…subunits of the oxidative phosphorylation (OXPHOS) that need to be assembled…” should be changed to “…subunits of the oxidative phosphorylation (OXPHOS) machinery that need to be assembled…” using machinery or a similarly descriptive term.

2.     Line 90: The inclusion of ‘subsection’ in the title “2.1. Mitochondrial dynamics Subsection” appears to be accidental as it does not align with the title naming conventions in the rest of the manuscript.

3.     Lines 151-153: The sentence spanning these lines should be split into separate sentences through changing “…we direct the readers to a recent review, in this work we review...” to “… we direct the readers to a recent review. In this work we will review…”.

4.     Line 53: “…generate accumulation…” should be changed to “…generate an accumulation…”; Line 91: “…functioning…” should be corrected to “…functions…”; Line 94: “…mitochondrial…” should be corrected to “…mitochondria…”; Line 124: “…targeting mitochondria to selective autophagy…” should be changed to “…targeting mitochondria for selective autophagy…”; Line 181: ‘the’ should be deleted in the phrase “…restoring the mitochondrial homeostasis…”; Line 256, line 305: the use of ‘increment’ should be changed to ‘increase’ or ‘decrease’ where relevant; Line 260: “By performing RNAi screen…” should be changed to “By performing an RNAi screen…”; Line 284: ‘t’ should be deleted after “…induce the UPRmt t…”.

5.     Line 318: ‘ETC’ should be defined in its first use in line 318 rather than in line 387.

Author Response

We thank the Reviewer for appreciating our work and effort. Comments to the specific points raised by the reviewer follow in red below. 

Main comments:

Point 1:     In section 2.2, the PINK-1 protein is occasionally referred to as PINK1 and the PDR-1 protein is sometimes referred to as Parkin. Consistent use of a single naming convention would help readers better follow along with the authors’ discussions. As the text is specifically discussing mitophagy in C. elegans, it would perhaps be beneficial to adopt the PINK-1/PDR-1 nomenclature that is unique to C. elegans.

We have now adopted the C. elegansnomenclature all through the text.

Point 2.     In the 4th paragraph of section 2.2, it is not clear from the text what the genetic background is of the “insulin mutants” and the “mitochondrial defective mutants” referred to by the authors. 

It is also not very clear what “hypoxia response” is referring to in this section.

It could also be helpful for readers if the authors provided some speculation on the relevance to mitophagy regulation and lifespan extension of the SKN-1/DCT-1 changes observed in the insulin mutant lines specifically mentioned in the text. 

We have now specified the alleles of the specific insulin daf-2(e1370)and mitochondrial mutants isp-1(qm150)and clk-1(e2519).

Regarding the hypoxia response, we referred to the fact that infection of Pseudomonas induces the expression of genes that are normally induced under hypoxia conditions. Since it is not relevant within the scope of the review, we have decided to remove it for clarity.

Point 3.     Section 2.3: The nuclear translocation of ATFS-1 in response to a disruption in its normal mitochondrial import has been well established as a central regulatory component of the UPRmt. Additionally, while there is not yet direct evidence showing the peptides released by HAF-1 interacting with mitochondrial import complexes, the actions of the HAF-1 transporter have been shown to cause a global repression in mitochondrial import which is critical for initiating the UPRmt through ATFS-1 (see PMID: 22700657 and PMID: 20188671 for both these points). While the authors do briefly mention that the nuclear import of ATFS-1 in response to disrupted mitochondrial protein import leads to the expression of UPRmt genes, and that peptides released by HAF-1 lead to ATFS-1 translocation to the nucleus, these points are mentioned in separate paragraphs and a connection between these two observations is not made clear in the text. As these series of events have been robustly shown to be central components of the UPRmt signalling upstream of ATFS-1 and DVE-1/UBL-5 mediated gene transcription, the authors’ description of UPRmt regulation in section 2.3 of the manuscript should be amended to reflect the importance of disrupted mitochondrial import of ATFS-1 to UPRmt induction.

Additionally, the disruption to mitochondrial import and subsequent change in ATFS-1 localisation are not referenced in Figure 1. Both the figure legend and the graphical components of Figure 1 should be updated to include descriptions of these events to more accurately represent the literature available on UPRmt signalling.  

We do appreciate the reviewer´s suggestion. We have now accordingly modified the figure, the figure legend and added a sentence in line 234 to clarify that impaired protein import into the mitochondria contributes to the nuclear translocation of ATFS-1. 

Minor comments:

1.     Line 52: The sentence “…subunits of the oxidative phosphorylation (OXPHOS) that need to be assembled…” should be changed to “…subunits of the oxidative phosphorylation (OXPHOS) machinery that need to be assembled…” using machinery or a similarly descriptive term.

We have now included “machinery” after OXPHOS, now in line 56.

2.     Line 90: The inclusion of ‘subsection’ in the title “2.1. Mitochondrial dynamics Subsection” appears to be accidental as it does not align with the title naming conventions in the rest of the manuscript.

We are thankful to the reviewer for spotting that mistake, which indeed was accidental. We have removed it.

3.     Lines 151-153: The sentence spanning these lines should be split into separate sentences through changing “…we direct the readers to a recent review, in this work we review...” to “… we direct the readers to a recent review. In this work we will review…”.

Now in line 191 we have split the sentence.

4.     Line 53: “…generate accumulation…” should be changed to “…generate an accumulation…”; 

Now in line 57 we have changed to ”generate an accumulation”

Line 91: “…functioning…” should be corrected to “…functions…”; 

This is now in line 94 and it has been corrected

Line 94: “…mitochondrial…” should be corrected to “…mitochondria…”; 

It has been corrected and is now line 97

Line 124: “…targeting mitochondria to selective autophagy…” should be changed to “…targeting mitochondria for selective autophagy…”; 

That has been done and is now line 151

 Line 181: ‘the’ should be deleted in the phrase “…restoring the mitochondrial homeostasis…”; Removed from now increaseline 237 

Line 256, line 305: the use of ‘increment’ should be changed to ‘increase’ or ‘decrease’ where relevant; 

Now in line 334 it reads: ”an increase of preteasomal activity”

Now in line 403 it reads: “lifespan enhancement upon mitochondrial…”

Line 260: “By performing RNAi screen…” should be changed to “By performing an RNAi screen…”; 

Done, now in line 344

Line 284: ‘t’ should be deleted after “…induce the UPRmt t…”.

Done, now in line 379

5.     Line 318: ‘ETC’ should be defined in its first use in line 318 rather than in line 387.

ETC is now for the first time defined in line 55. Additionally, we added it also in the Figure 2 legend.

We hope the reviewer considers the modifications satisfactory. 

Reviewer 3 Report

Hernando-Rodriguez and Artal-Sanz review the literature regarding mitochondrial quality control and stress response mechanisms with a focus on the model organism Caenorhabditis elegans.

The manuscript covers this broad topic comprehensively, the subsections are well chosen, and the figures nicely illustrate key aspects of the manuscript.

I am convinced the manuscript will be of high interest to the readership of Cells.

Parts of the manuscript are not easy to follow and sometimes confusing, because single aspects have not been sufficiently introduced. It seems as if the order of the chapters in the manuscript has possibly been rearranged at some point, without applying all necessary text changes, and the manuscript would benefit from a very thorough round of proofreading with special focus on these points. In order to make the manuscript more accessible to readers, I furthermore suggest the following minor changes:

1.)   The authors use abbreviations very extensively, and quite inconsistently. Sometimes, abbreviations are not being introduced when the term is mentioned for the first time, but at some point later, are being introduced multiple times, or are being introduced although the term is only used once in the whole manuscript. Especially because the topic is very broad and most readers will likely not be experts in all subtopics covered, this can make it difficult to follow the logic of the manuscript. I suggest to reduce abbreviations to an absolute minimum, and to take extra care to introduce the ones that are being used carefully and correctly.

2.)   The manuscript requires proofreading by a native speaker.

3.)   The following points have to be corrected:

a.     L14: “import of thousands of proteins”. In most species, the number of known mitochondrial proteins is below 2000.

b.     L35: “Ca2+ homeostasis and Fe-S cluster biogenesis, two processes involved in health”. This is a surprising statement, as it appears likely that most cellular processes contribute to health.

c.     L52: include “machinery” after OXPHOS

d.     L78: “we will revise the main mitochondrial stress response pathways”. Rather “review the main mitochondrial stress response pathways”?

e.     L101: The sentence should start with “In C. elegans,…”, because this is the first time the review focusses on worm, which is not clear from the text at this point.

f.      L134-148: Insulin mutants require a few words of introduction. What does “both” transcription factors refer to? Should “Increased lethality to Pseudomonas treatment” rather be “Increased lethality upon Pseudomonas infection”?

g.     L213: What does “both phenotypes” refer to?

h.     L262: clk-1 mutants require introduction

i.      L401: Is Tim14p supposed to be the yeast name? If yes, please use Pam18p, which is the official name (Tim14p is an alias).

j.      L492: “TIM and PAM complexes that function to transport proteins into the inner membrane”: please change to “into or across”, because most TIM/PAM substrates are soluble matrix proteins, while only a smaller subset are inner membrane proteins.

Author Response

We appreciate the reviewers encouraging comments. We have read carefully the manuscript putting attention focusing on the points mentioned by the reviewer. We have also added paragraphs as per the reviewer´s suggestions and hope the manuscript can be now followed easily.

I furthermore suggest the following minor changes:

1.)   The authors use abbreviations very extensively, and quite inconsistently. Sometimes, abbreviations are not being introduced when the term is mentioned for the first time, but at some point later, are being introduced multiple times, or are being introduced although the term is only used once in the whole manuscript. Especially because the topic is very broad and most readers will likely not be experts in all subtopics covered, this can make it difficult to follow the logic of the manuscript. I suggest to reduce abbreviations to an absolute minimum, and to take extra care to introduce the ones that are being used carefully and correctly.

We have now very carefully checked the abbreviations used and that they are properly introduced to the reader when mentioned for the first time (see also response to reviewer 2 comments)

2.)   The manuscript requires proofreading by a native speaker.

The manuscript has now been read proofread by a native speaker.

3.)   The following points have to be corrected:

a.     L14: “import of thousands of proteins”. In most species, the number of known mitochondrial proteins is below 2000.

We have now written “a large number of proteins” hoping the reviewer will it find appropriate.

b.     L35: “Ca2+ homeostasis and Fe-S cluster biogenesis, two processes involved in health”. This is a surprising statement, as it appears likely that most cellular processes contribute to health.

We have now removed “involved in health” and exchanged by “essential cellular processes”

c.     L52: include “machinery” after OXPHOS

We have included “machinery” after OXPHOS, now in line 56

d.     L78: “we will revise the main mitochondrial stress response pathways”. Rather “review the main mitochondrial stress response pathways”?

Now in line 81. In order to avoid repeating “review” as the sentence was “in this review, we will revise…” we have now stated: “Herein, we will review the main mitochondrial stress response pathways”

e.     L101: The sentence should start with “In C. elegans,…”, because this is the first time the review focusses on worm, which is not clear from the text at this point. 

Now in line 115, we start the sentence as suggested by the reviewer. In C. elegans, mitochondrial fission…and the paragraph has been extended as per the suggestion of reviewer 1. 

f.      L134-148: Insulin mutants require a few words of introduction. 

Now lines 175. Following also comments by Reviewer 2, now this paragraph includes description and specific allele of the insulin mutant we referred to. 

What does “both” transcription factors refer to? 

We have changed to “two transcription factors”, which are mentioned right after “DAF-16 and SKN-1”.

Should “Increased lethality to Pseudomonas treatment” rather be “Increased lethality upon Pseudomonas infection”?

We thank the reviewer for this correction which we have implemented in line 182.

g.     L213: What does “both phenotypes” refer to?

Line 274: We apologise for this mistake and have now changed to “to induce the UPRmt”; we were refereeing only to the UPRmt.

h.     L262: clk-1 mutants require introduction

A brief introduction to clk-1 has now been included in line 127

i.      L401: Is Tim14p supposed to be the yeast name? If yes, please use Pam18p, which is the official name (Tim14p is an alias).

We thank the reviewer for this remark, we have now called it Pam18p. Now in Line 500

We have also changed the figure legend accordingly (Line 466) 

j.      L492: “TIM and PAM complexes that function to transport proteins into the inner membrane”: please change to “into or across”, because most TIM/PAM substrates are soluble matrix proteins, while only a smaller subset are inner membrane proteins.

We appreciate this correction made by the reviewer and have changed it to “into or across” in Line 593.

We hope the reviewer finds the revised version of this manuscript suitable for publication.